# Personal Relative Deprivation and Online Aggression in College Students: A Moderated Mediation Model of Revenge Motivation and a Violent Attitude

**DOI:** 10.3390/bs14111108

**Published:** 2024-11-18

**Authors:** Wenfeng Zhu, Yuguang Yang, Xue Tian, Yongchao Huang, Xuejun Bai

**Affiliations:** 1Key Research Base of Humanities and Social Sciences of the Ministry of Education, Academy of Psychology and Behavior, Tianjin Normal University, Tianjin 300387, China; yyg1159514878@163.com (Y.Y.); tx19900109@126.com (X.T.); 2Faculty of Psychology, Tianjin Normal University, Tianjin 300387, China; 3Tianjin Key Laboratory of Student Mental Health and Intelligence Assessment, Tianjin 300387, China; 4Tianjin Jinghai Experimental School Affiliated to Beijing Normal University, Tianjin 300387, China; huangyongchao521@163.com

**Keywords:** personal relative deprivation, online aggression, violent attitude, revenge motivation, social comparisons, individual differences

## Abstract

While personal relative deprivation (PRD) is recognized as a potential risk factor for aggression, the mechanisms underlying this relationship are not well understood. This study investigates how revenge motivation mediates the link between PRD and online aggression, as well as how a violent attitude moderates this connection. A total of 1004 college students completed self-reported measures on demographic factors, PRD, online aggression, revenge motivation, and violent attitudes. The findings revealed a positive correlation between PRD and online aggression, with revenge motivation serving as a mediating factor. Additionally, a violent attitude was found to moderate the relationship, indicating that PRD had a stronger association with online aggression in individuals with higher violent attitudes compared to those with lower attitudes.

## 1. Introduction

As of December 2021, the number of internet users in China reached 1.032 billion, and the internet penetration rate reached 73.0% [1]. For today’s youth, the Internet and digital media are essential components of everyday life [2]. Among college students, the Internet presents both advantages and disadvantages. On the one hand, it facilitates communication and knowledge sharing. For instance, the Internet has created new opportunities for students to find online support for their academic difficulties [3]. Utilizing social media platforms like Facebook, students can seek assistance from peers and instructors or look for pertinent information online to effectively address their academic needs [4]. Conversely, internet usage can lead to negative issues, such as online aggression [5]. A study on cyberbullying among Chinese college students revealed that roughly 39.18% have participated in online aggression while online [6].

Online aggression represents a novel form of aggression that occurs through the Internet, characterized by repeated aggressive actions over time via electronic devices such as social networking platforms and email. This issue is becoming increasingly prevalent among college students [7,8,9]. Unlike traditional aggression that occurs in face-to-face interactions, online aggression is marked by a lack of time and spatial constraints, extensive impact, potential for repeated harm, and significant anonymity [10,11,12,13]. Anonymity, known as unidentifiability, also referred to as identifiability, is the degree to which users feel their real names or true identities can be concealed in a channel [14]. Some research underscores the pivotal role that anonymity plays in facilitating online aggression [15,16]. Theoretical approaches such as SIDE suggest that anonymity drives feelings of depersonalization and disinhibited behavior [17]. Additionally, anonymity often leads bullies to believe that they can evade punishment or retaliation following an online attack [18]. In the cultural context of China, emotional repression is viewed as a behavior that aligns with traditional values, and individuals tend to utilize expressive suppression more frequently in emotion regulation, which is likely rooted in Confucian values that promote self-restraint, moderation, and harmonious interpersonal relationships [19]. The anonymity and convenience of the Internet may lead individuals who feel deprived to resort to online aggression as a way to vent their repressed emotions and restore balance. Numerous studies have indicated that online aggression can result in serious negative effects for victims, including anxiety, depression, academic failure, Internet addiction, substance abuse, and even suicide [20,21,22,23,24,25]. Therefore, it is essential to explore the factors and psychological mechanisms influencing college students’ online aggression in order to effectively prevent and intervene in this behavior.

### 1.1. Personal Relative Deprivation and Online Aggression

PRD is a profound emotional and psychological phenomenon rooted in the erosion of happiness caused by persistent inequalities, such as economic disparities. This concept encapsulates the sentiments of resentment and discontentment that stem from an individual’s perception of being unfairly deprived of outcomes they perceive as their due, in comparison to a reference group [26]. Some studies have found that PRD is a significant risk factor associated with online aggression [27,28]. The PRD theory posits that individuals or groups become aware of their disadvantages by comparing themselves either horizontally or vertically to a reference group, leading to negative emotions such as depressive symptoms [29,30,31,32] and anger [33,34,35], which can heighten the likelihood of aggressive behavior [36,37,38]. Additionally, fairness theory suggests that when individuals perceive unfair treatment, they not only feel dissatisfaction but also alter their behavior to restore a sense of fairness [39]. Specifically, due to the anonymity and ease of access provided by the Internet, individuals experiencing feelings of deprivation may resort to online aggression as a means to express their negative feelings and regain equilibrium.

Prior research has demonstrated a positive correlation between PRD and various forms of aggressive behavior [40,41,42], antisocial behaviors [43,44], and criminal behavior [42,45]. However, most studies have focused on the effects of PRD on conventional aggressive behaviors [46,47], with comparatively less emphasis on its connection to online aggression. Furthermore, the specific mechanisms and timing through which PRD influences online aggression remain unclear. This study aims to address this gap by investigating whether revenge motivation acts as a mediator in the relationship between PRD and online aggression, as well as exploring whether a violent attitude serves as a moderator for both the direct and indirect effects.

### 1.2. Revenge Motivation as a Mediator

Revenge motivation refers to why an individual hopes to retaliate or harm the offender due to righteous indignation after being provoked [48]. We assume that revenge motivation may play a mediating role in the association between PRD and online aggression.

On the one hand, the theory of relative deprivation holds that PRD will lead to an individual’s sense of injustice, which not only fosters feelings of anger but also generates revenge motivation. Specifically, the sense of deprivation often triggers feelings of injustice and frustration, which can create a perceived threat to one’s social standing or identity. This emotional response can then translate into a desire for revenge as individuals seek to restore their sense of fairness and control [31,49,50]. A previous study showed that the injustice perception triggered by PRD increases people’s motivation to retaliate against perpetrators [51]. Moreover, Hu (2016) also found that inequalities triggered by PRD could trigger revenge motivation in the workplace [52]. Therefore, we hypothesize that revenge motivation mediates the relationship between PRD and online aggression.

Conversely, revenge motivation is a predictor of online aggression. Research has established a positive link between revenge motivation and various serious crimes, including school shootings [53], genocide [54], rape [55], homicide/suicide [56], and civil war atrocities. The new information and communication technologies (ICTs) model indicated that revenge motivation is an important motivation underlying online aggression [57]. Ruions identified four types of network attacks, among which the impulsive–aversive and controlled–aversive types are directly related to retaliatory motives [57]. Nevertheless, revenge does not always escalate to such severe acts as school shootings or civil wars, making it worthwhile to investigate its influence on less severe and more prevalent forms of online aggression. Compton (2014) noted that individuals are more inclined to use online aggression as a safe way to retaliate [18]. His research revealed that both teachers and parents perceived the internet as a platform where individuals tend to express themselves more freely than in face-to-face interactions. One teacher likened cyberbullying to “smarter bullying,” suggesting that perpetrators who engage in this form of bullying feel a sense of protection due to their anonymity. Recent findings indicate that revenge motivation is positively correlated with online aggression in the context of adolescent romantic relationships [58]. Therefore, we assume that revenge motivation may positively predict online aggression.

Based on the literature review, we can speculate that PRD makes college students feel unfairly treated [34], which further leads to revenge motivation [59] and subsequently increases the likelihood of engaging in online aggression. In summary, revenge motivation acts as a mediating factor in the relationship between PRD and online aggression.

### 1.3. Violent Attitude as a Moderator

College students experiencing high levels of perceived PRD are more prone to engage in online aggression; however, this does not imply that all such students will exhibit high levels of aggressive behavior. In fact, college students with different evaluation and reaction tendencies to violence may respond differently to similar environmental contexts [60,61].

According to the general aggression model [62], the evaluation of the appropriateness of aggressive behavior influences whether or not aggression was utilized to solve problem. One of components of the evaluation is the judgment of whether aggressive behavior is in line with individuals’ self-regulating internal standers (e.g., internalized moral standers and social norms). When individuals perceive aggressive behavior as incongruent with social norms, aggressive behaviors are unlikely to be carried out [62,63]. A violent attitude is an important factor for the evaluation process and may play a critical role in reducing the association between PRD and online aggression. It refers to the tendency of an individual to evaluate and explain violence in a positive way [2,64]. When individuals suffer from PRD, they may choose whether to carry out aggressive behavior based on their own attitude to the aggression. Specifically, individuals with a highly violent attitude are prone to evaluating and interpreting violent behavior in a positive light [65]. These tendencies make them more likely to view violent behavior as acceptable and show more aggression than those with low violent attitudes [66]. Thus, a high level of violent attitude may exacerbate the association between PRD and online aggression.

### 1.4. Current Study and Hypotheses

In this study, we investigate whether revenge motivation acts as a mediator in the connection between PRD and online aggression, as well as whether a violent attitude serves as a moderator in this relationship (refer to Figure 1).

**Hypothesis** **1.**
*PRD can predict online aggression.*


**Hypothesis** **2.**
*Revenge motivation mediates the relationship between PRD and online aggression.*


**Hypothesis** **3.**
*A violent attitude moderates the relationship between PRD and online aggression.*


## 2. Materials and Methods

### 2.1. Participants

A total of 1009 Chinese college students were recruited from Guangxi Normal University, Yunnan Normal University, and Dezhou University. The questionnaires were administered online with assistance from research aides in each university class. After removing data from participants with incomplete information (5 participants did not complete the measures), a total of 1004 valid responses were retained, resulting in a retention rate of 99.5%. The average age of the participants was 19.65 years (SD = 0.834, range = 16–24 years), consisting of 262 males (26.1%) and 742 females (73.9%). The numbers of participants from three universities were 589, 235, and 180, respectively. The male-to-female ratios were as follows: 180 (30.6%)/409 (68.93%); 30 (16.66%)/150 (83.33%); and 50 (21.28%)/185 (78.72%). The ages of participants from the three universities ranged from 18 to 22 years old, with response rates of 99.49%, 100%, and 99.16%, respectively. This research received approval from the scientific research ethics committee at the institution of the first author. Informed consent was obtained from all participants prior to data collection, and they were assured that there were no right or wrong answers and that their data would be used solely for research purposes. Participants received RMB 15 as compensation for completing the survey.

### 2.2. Measures

All scales have been used for Chinese participants and have shown good reliability and validity.

#### 2.2.1. Personal Relative Deprivation

The study used the personal relative deprivation scale [67] to evaluate PRD perception among college students. The questionnaire includes five items, such as “I feel at a disadvantage compared to people like me”. Respondents rated each statement on a six-point scale from 1 (strongly disagree) to 6 (strongly agree). A higher total score indicates a greater perception of PRD. In this study, the Cronbach’s α was 0.916.

#### 2.2.2. Online Aggression

The study used the Cyber-Aggression Typology Questionnaire (CATQ) to measure the severity of online aggression [68]. The CATQ comprises 29 statements, such as “If someone tries to hurt me, I will immediately retaliate against them through the internet”. Each item is rated on a scale from 1 (strongly disagree) to 4 (strongly agree). In this study, the Cronbach’s α coefficient was 0.973.

#### 2.2.3. Revenge Motivation

Revenge motivation was assessed using the revenge dimension of the Transgression-Related Interpersonal Aggression Motivation Scale (TRIM) created by McCullough [48]. Participants were instructed to “please indicate your current thoughts and feelings about the person who recently hurt you”. Then, they needed to read five items, such as “I will pay him back”. Participants rated each item on a 5-point scale, where 1 meant “not at all like me” and 5 meant “very much like me”. A higher total score reflects a greater level of revenge motivation. In this study, the Cronbach’s alpha was 0.904.

#### 2.2.4. Violent Attitude

The Attitudes Toward Violence Scale (ATVAS) was used to assess violent attitude [69]. It consists of 14 items, and the college students rated each statement, such as “If someone hits you, you should hit back,” on a five-point scale ranging from 1 (strongly disagree) to 5 (strongly agree). Higher scores indicate a greater propensity for violent behavior in various situations. In this study, the Cronbach’s α was 0.831.

### 2.3. Procedure

The study adhered to the principles outlined in the Declaration of Helsinki. Ethical approval was granted by the Ethics Committee for Scientific Research at the university of the first researcher. Participants filled out self-report questionnaires after being briefed on the study’s purpose and content. They were assured that the survey would remain anonymous and confidential, and they had the right to withdraw from the study at any point.

### 2.4. Data Analysis

First, we calculated descriptive statistics and Pearson correlations using SPSS 24.0. Next, we conducted structural equation modeling (SEM) with Mplus 8.3 to investigate the mediation effect. Finally, we examined whether the mediation process was moderated by violent attitudes using Hayes’ PROCESS macro (Model 5). To assess model fit, we employed various goodness-of-fit indices as recommended by Hu and Bentler (1999) [70]. The criteria for acceptable fit included a Root Mean Square Error of Approximation (RMSEA) of 0.08 or lower, a Standardized Root Mean Square Residual (SRMR) of 0.08 or lower, and a Comparative Fit Index (CFI) greater than 0.90. Additionally, we utilized a bootstrap procedure to evaluate the size of the indirect effect of PRD and its confidence intervals (CI). In addition, given the significant gender differences in relative deprivation, online aggression, and violent attitudes, we used Model 8 and Model 14 to explore gender as a moderator in the mediation relationship. Model 3 was used to examine the moderating effects of gender and violent attitudes in the relationship between relative deprivation (PRD) and online aggression.

## 3. Results

### 3.1. Preliminary Analyses

Table 1 provides the means, standard deviations, and correlations for the key variables. Gender was coded as a dummy variable, with 1 for male and 2 for female. The findings revealed that males reported higher levels of PRD, more violent attitudes, and greater online aggression than females, with significant correlations noted *(t* = 2.385, *p* = 0.015; *t* = 7.668, *p* < 0.001; *t* = 5.571, *p* < 0.001). However, no significant gender differences were observed in TRIM (*t* = −0.693, *p* = 0.488). Additionally, the four primary variables were found to be correlated with one another.

### 3.2. Testing for Mediation Effect

We utilized latent moderated structural equations to explore the mediating effect of revenge motivation on the relationship between PRD and online aggression. The mediation model, which included revenge motivation as a mediator, demonstrated a good fit to the data: χ^2^/df = 4.715, CFI = 0.988, TLI = 0.982, RMSEA = 0.061, and SRMR = 0.025. The direct path coefficient from PRD to online aggression was significant (*β* = 0.238, *p* < 0.001).

To evaluate the size of the indirect effect and its confidence intervals (CI), we employed a bootstrap procedure, generating 5000 bootstrap samples through random sampling from the original dataset. The analysis showed an indirect effect of 0.130 (SE = 0.006, 95% CI = [0.019, 0.042]). Since the 95% CI did not include zero, it confirmed that the indirect effect of revenge motivation was significant, indicating that revenge motivation mediated the relationship between PRD and online aggression. The results are shown in Figure 2.

### 3.3. The Moderating Effect of a Violent Attitude

We explored whether the mediation process was moderated by a violent attitude using Hayes’ (2013) PROCESS macro (Model 5) [71]. The results, as shown in Table 2, showed that a violent attitude has a negative impact on college students’ online aggression (*β* = 0.402, 95% CI = [0.350, 0.455], *p* < 0.001). The prediction effect of the interaction term of PRD and a violent attitude on college students’ online aggression was significant (*β* = 0.076, 95% CI = [0.040, 0.112], *p* < 0.001), indicating that a violent attitude can moderate the relationship between PRD and online aggression. In summary, the link between PRD and online aggression is not only mediated by revenge motivation, but also moderated by a violent attitude. To further analyze the moderating effect of a violent attitude on the association between PRD and online aggression, violent attitude was divided into high (M + 1SD) and low groups (M − 1SD), and a simple slope test was performed (see Figure 3). For the individuals with high levels of violent attitude, PRD could significantly predict online aggression (*β*_simple_ = 0.105, *p* < 0.001), while the relationship was not significant for individuals with a low violent attitude (*β*_simple_ = 0.021, *p* = 0.162).

### 3.4. The Moderating Effect of Gender

Based on the significant differences observed in relative deprivation, online aggression, and violent attitudes between genders, we employed Model 8 and Model 14 to explore the role of gender as a moderator in the mediation relationship. Additionally, Model 3 was utilized to assess the moderating effect of gender on violent attitudes in the context of the relationship between the independent variable, PRD, and the dependent variable, online aggression.

The findings reveal that gender significantly moderates the relationship between revenge motivation and online aggression (*β* = −0.079, 95% CI = [−0.143, −0.015], *p* < 0.05). A simple slope analysis (see Figure 4) indicated that, for individuals with high levels of revenge motivation, gender significantly predicts online aggression (*β*_simple_ = 0.158, *p* < 0.001). This relationship remains significant for individuals with low levels of revenge motivation as well (*β*_simple_ = 0.079, *p* < 0.001).

## 4. Discussion

Although the relationship between PRD and traditional aggression has significant empirical support, the relationship between PRD and online aggression and the mediating and moderating mechanisms behind this relationship are still largely unknown. Therefore, we used a moderated mediation model to test whether revenge motivation mediates the relationship between PRD and college students’ online aggression and whether a violent attitude moderates the relationship between PRD and online aggression. The results show that PRD positively predicted online aggression among college students, with revenge motivation serving as a partial mediator in this relationship, suggesting that PRD retains a direct effect on online aggression. Moreover, violent attitudes moderated the relationship between PRD and online aggression. The result deepens our understanding of the relationship between PRD and online aggression.

### 4.1. The Mediating Role of Revenge Motivation

Consistent with our expectations, revenge motivation mediated the relationship between PRD and online aggression. First, individuals with feelings of PRD were more likely to engage in online aggression [46,47]. The result provides empirical support for fairness theory. The theory emphasizes that when people experience injustice, they not only become dissatisfied, but also adjust their behavior to restore fairness [39]. This result is also consistent with Tobias’s study, which indicated that individuals with low subjective socioeconomic status are more likely to feel disadvantaged, leading to a sense of relative deprivation, which in turn prompts them to exhibit higher levels of aggression [41]. Given the opportunity, individuals who experience PRD will take revenge on the source of their sense of relative injustice.

Secondly, PRD can predict revenge motivation. According to the theory of relative deprivation, the reason for revenge motivation may be the individual’s feeling of unfairness caused by the individual’s relative deprivation [31,49,50]. As demonstrated in a previous study related to revenge motivation, PRD increases people’s motivation to retaliate against perpetrators [51]. Moreover, individuals who experience a sense of PRD are more likely to have revenge motivation, for example, a workplace-related study found that PRD-induced inequalities may trigger retaliatory motives in the workplace [31,49,50,52].

Third, our result reveals that revenge motivation can predict online aggression, which is in line with previous studies. These studies showed that individuals’ revenge motivation is one of the predisposition factors inducing online aggression [72,73], and the result also provides empirical support for the (ICT) model. In addition, a recent study mentioned that revenge motivation is positively correlated with online aggression in adolescents’ romantic relationships [58]. Therefore, the generation of revenge motivation drives individuals to address PRD through online aggression behavior [18].

### 4.2. The Moderating Role of a Violent Attitude

Our findings suggest that a violent attitude moderates the relationship between PRD and online aggression. Specifically, compared with students with low violent attitudes, college students with high violent attitudes have a stronger connection between PRD and online aggression. These findings can be explained by the Moderating Function Theory of Attitude, which suggests that individuals may choose to engage in aggressive behavior based on their attitudes toward aggression [74]. College students with high violent attitudes are more inclined to evaluate and explain violent behavior positively. They are more likely to choose aggressive behavior to solve problems. Conversely, individuals with low violent attitudes disapprove of solving problems with aggressive behavior, so they will not choose to solve problems with aggressive behavior even if they feel PRD. In other words, compared with college students with low violent attitudes, individuals with high violent attitudes are more likely to commit online aggression against others when they experience PRD. Therefore, PRD has a significant predictive effect on online aggression for individuals with high violent attitudes. In contrast, for individuals with low violent attitudes, the predictive effect of PRD on online aggression is weakened.

### 4.3. The Moderating Role of Gender

Using gender as a moderating variable, we examined its effects on the mediating pathways and attitudes toward violence. The findings reveal that gender significantly moderates the relationship between revenge motivation and online aggression. Specifically, male college students demonstrate stronger associations between revenge motivation and online aggression compared to their female counterparts. This observation aligns with existing literature indicating that males tend to exhibit higher levels of aggression than females [75,76].

### 4.4. Limitations and Contributions

Several limitations need to be considered in the current study. First of all, all variables were assessed by self-reporting, which may affect this study’s validity [77]. Future research could collect data from multiple informants (such as parents, teachers, and peers) or experimental methods to reduce the influence of subjectivity. Secondly, the measurement of violent attitudes was based solely on self-reporting. There are well-known inherent problems, like the effect of stigmatization or the lack of introspection among some participants [78]. Therefore, future research can be utilize behavioral experiments or neuroscience to observe the psychological characteristics of reactions.

Thirdly, our study sample primarily consists of university students in China, a specific population that may be profoundly influenced by Chinese culture. This culture emphasizes collectivism, harmony, and the concept of “face”, which may lead individuals in China to be more inclined to express their emotions through online aggression. In situations where direct conflict or dissatisfaction is challenging to articulate, online platforms provide a relatively concealed means of venting negative emotions. As a result, the applicability and generalizability of our findings may be limited, particularly when considering other countries and cultural contexts. Future research should consider samples from different cultural backgrounds to validate and expand our findings, thereby achieving a more comprehensive understanding of the diversity in online behavior. Fourthly, our study on online aggression primarily focuses on various behaviors and feelings individuals may experience when interacting with others through platforms such as text messages, emails, online chat rooms, and more. However, for other forms of online aggression, the results could differ. Future research could consider incorporating additional forms of online aggression to verify the stability of the results. Fifth, the current study is only cross-sectional, which can only point out associations between variables, not causation. Therefore, future studies could use a longitudinal design to explore the relationship between PRD and online aggression. Finally, this study only focused on the individual cognitive variables of the relationship between PRD and online aggression and did not examine interpersonal or environmental variables. Therefore, in future studies, other personality dimensions, individual psychopathology, interpersonal relationships, family environment, and other variables should be included to enrich our understanding of the relationship between PRD and online aggression [79].

Despite these limitations, compared with simple mediating or moderating models, this study provides deeper insight into the internal mechanism of the influence of PRD on online aggression, which has specific theoretical and practical significance and provides enlightenment for preventing online aggression among college students. Theoretically, this study is the first to investigate the relationship between PRD and online aggression and confirms the mediating effect of revenge motivation and the moderating effect of a violent attitude. Our findings extend previous research on online aggression and suggest why and how PRD predicts online aggression. In practice, our study may provide some help for the prevention and intervention of online aggression. First of all, PRD can predict online aggression through revenge motivation, which means educators should channel the PRD of college students promptly to reduce revenge motivation in order to avoid online aggression in turn. We also recommend the following strategies to address retaliation-driven online aggression. First, educational institutions should prioritize the development of students’ emotional intelligence and conflict resolution competencies [80]. Online awareness campaigns on social media should highlight the consequences of aggression while encouraging positive behaviors [81]. Establishing support networks will provide safe spaces for sharing experiences and developing healthy coping strategies [82]. Collaborating with social media influencers can amplify messages of kindness, and advocating for clearer platform guidelines will help deter aggressive behaviors. Together, these strategies aim to foster a more positive online environment. In addition, PRD was more strongly associated with online aggression among individuals with high violent attitudes than those with low violent attitudes. Educators can attenuate the effects of PRD on the relationship between online aggression by fostering proper attitudes towards violence among college students.

## 5. Conclusions

This study has significantly contributed to our understanding of the intricate relationship between PRD, revenge motivation, violent attitude, and online aggression among college students. By employing a moderated mediation model, we have confirmed that revenge motivation serves as a crucial mediating variable between PRD and online aggression, elucidating the psychological mechanisms underlying this phenomenon. Furthermore, we have demonstrated that violent attitude moderates the direct and indirect relationships between PRD and online aggression, revealing that individuals with high violent attitudes are more susceptible to engaging in online aggression when experiencing PRD. Finally, we also found that gender moderates the relationship between revenge motivation and online aggression. These findings not only advance our theoretical knowledge, but also have practical implications for preventing and intervening in online aggression among college students. Educators and policymakers can leverage these insights to develop targeted interventions that address revenge motivation and foster more positive attitudes towards violence, thereby mitigating the adverse effects of PRD on college students’ online behavior.

## Figures and Tables

**Figure 1 behavsci-14-01108-f001:**
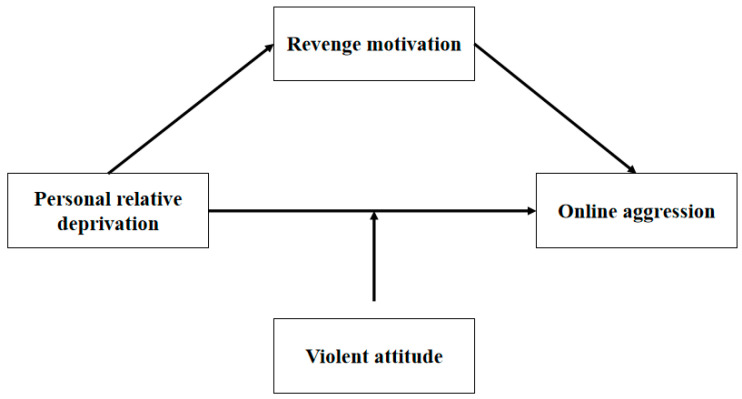
Proposed moderated mediation model.

**Figure 2 behavsci-14-01108-f002:**
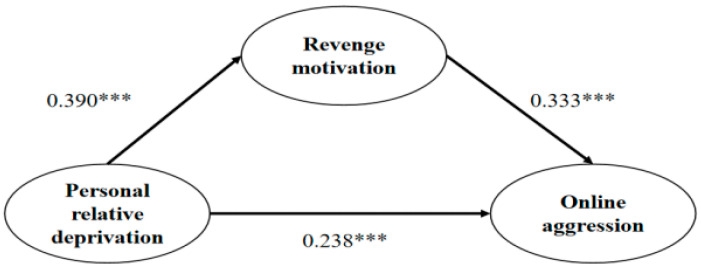
The mediation model with standardized path coefficients *** *p* < 0.001.

**Figure 3 behavsci-14-01108-f003:**
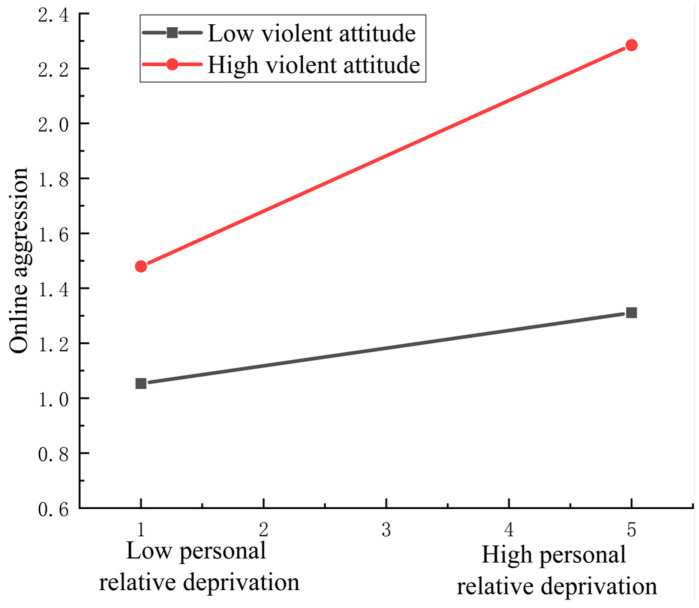
Online aggression as a function of PRD and violent attitude. Functions are graphed for two levels of violent attitude: 1 standard deviation above the mean and 1 standard deviation below the mean. Note that the graph is for descriptive purposes only.

**Figure 4 behavsci-14-01108-f004:**
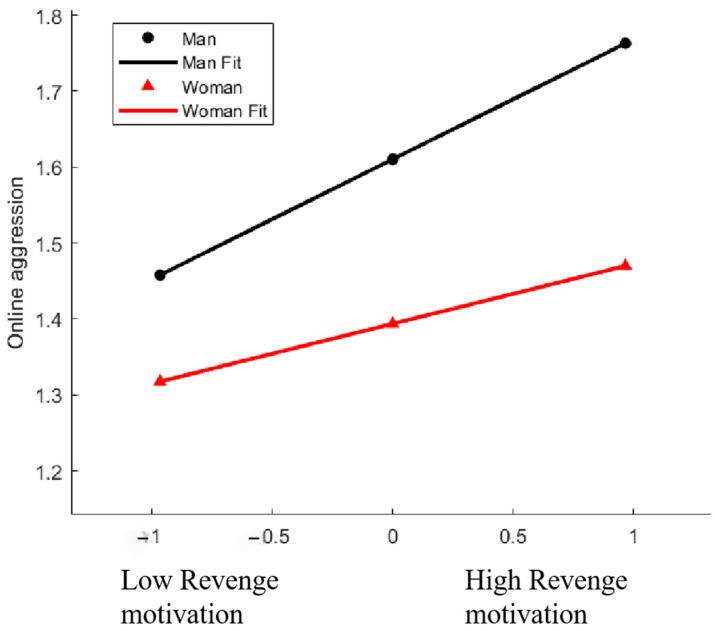
Online aggression as a function of revenge motivation and gender. Functions are graphed for two levels of gender: 1 standard deviation above the mean and 1 standard deviation below the mean. Note that the graph is for descriptive purpose only.

**Table 1 behavsci-14-01108-t001:** Descriptive statistics and correlations among variables (*N* = 1004).

	M ± SD	1	2	3	4	5	6
Gender	−	1					
Age	19.65 ± 0.83	−0.026	1				
PRD	2.91 ± 1.10	−0.075 *	0.002	1			
TRIM	2.69 ± 0.99	0.022	−0.051	0.376 **	1		
ATVAS	1.92 ± 0.53	−0.235 **	0.023	0.324 **	0.389 **	1	
CATQ	1.45 ± 0.49	−0.173 **	0.001	0.352 **	0.405 **	0.590 **	1

Note: SD = standard deviation, PRD = personal relative deprivation, TRIM = revenge motivation, ATVAS = violent attitude, CATQ = online aggression. ** *p* < 0.01, * *p* < 0.05.

**Table 2 behavsci-14-01108-t002:** Moderated mediation effect analysis of the relationship between PRD and online aggression.

	Model 1 (Revenge Motivation)			Model 2 (Online Aggression)		
Predictors	*β*	t		*β*	t	
Gender	0.111	1.677		−0.066	−2.329 *	
Age	−0.060	−1.732		−0.001	−0.083	
PRD	0.341	12.966 *		0.061	5.102 **	
Va				0.402	14.943 **	
PRD × Va				0.076	4.195 **	
R2			0.147			0.412
F			57.228 **			116.447 **

Gender was dummy-coded such that 1 = female and 0 = male. Va = violent attitude. PRD = personal relative deprivation. ** *p* < 0.01, * *p* < 0.05.

## Data Availability

The datasets generated during and/or analyzed during the current study are available from the corresponding author upon reasonable request.

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
