# Peer review of "Personal Relative Deprivation and Online Aggression in College Students: A Moderated Mediation Model of Revenge Motivation and a Violent Attitude"

_behavsci, 2024, doi:10.3390/bs14111108_

Round 1
Reviewer 1 Report
Comments and Suggestions for Authors
Comments
1. Introduction, page 1: Since this research was not conducted during COVID-19, I would encourage the authors not to list only remote classes during COVID-19 as an advantage for internet use. I suggest mentioning a more present and permanent advantage (e.g., other communication and knowledge sharing).
2. Introduction, page 1: The authors should explain the affordances that media provides, which makes aggression more prominent in online environments among Chinese students. The authors briefly mention anonymity, but this should be emphasized more as it is essential in this context. I recommend reading the following article for clarification: Fox, J. & McEwan, B. (2017). Distinguishing technologies for social interaction: The perceived social affordances of communication channels scale, Communication Monographs, 84(3), 298-318, DOI: 10.1080/03637751.2017.133241
3. Personal relative deprivation and online aggression: The text reads, “Personal relative deprivation (PRD) is a significant risk factor associated with online aggression.” However, there is no citation and no mention of any studies on PRD and online aggression. This should be cited, and studies related to online aggression should be included. If there are no studies, this sentence is not supported.
4. A definition of PRD should be provided. Also, after the first time spelling out what “PRD” stands for, the authors should use the abbreviation.
5. I recommend explaining the following sentence a bit more: “…use online aggression as a safe way to retaliate.” (line 83, page 2). What does “a safe way to retaliate” imply? How does this relate to your sample and the use of aggression online versus offline?
6. Current study and hypothesis: Why wasn’t “a violent attitude” included as a moderator of the link between revenge motivation and online aggression as well?
7. Materials and Methods: Was this paragraph on page 3 supposed to be deleted? It is an instruction to authors rather than to readers. “The Materials and Methods section should provide enough detail to enable others to replicate and expand upon the published findings. By submitting your manuscript, you agree to make all associated materials, data, computer code, and protocols accessible to readers. At the time of submission, please indicate any limitations regarding the availability of these materials or information. New methods and protocols should be explained comprehensively, while established methods can be summarized with appropriate citations.”
8. Participants: How many participants were from each University? Where samples similar to each other in terms of age, sex, etc.? What was the response rate?
9. Procedure: Did participants provide written consent?
10. Data Analysis: Why was the SEM conducted in Mplus, but the correlations and assumably Hayes’ PROCESS conducted in SPSS? I am confused as to why the mediation was done using SEM, but the moderation was done using PROCESS.
11. On line 231, it reads, “…while the relationship becomes weaker for individuals with a low violent attitude (bsimple = 0.024, p = 0.162).” I would suggest saying that the relationship was not significant rather than that it becomes weaker as this sometimes implies that it is still significant.
12. On page 8, line 261, the text reads, “….revenge motivation mediated this relationship.” I think it is worth noting that it was a partial mediation, suggesting that PRD still directly predicted online aggression.
13. This sentence (page 8, line 271) could be moved up to the introduction to provide support for your study “…due to the anonymity and convenience of the Internet, when people feel deprived, they may resort to online aggression to vent negative experiences and restore balance [65].”
14. This sentence does not provide any information (page 8, line 274); please tie the study to your findings: “This result is also consistent with Tobias's study [32].”
15. I recommend moving introductions and explanations of theories (equity theory) and models (ICT) to the introduction.
Reviewer 2 Report
Comments and Suggestions for Authors
Thank you for submitting your manuscript titled "Personal Relative Deprivation and Online Aggression in College Students: A Moderated Mediation Model of Revenge Motivation and a Violent Attitude" to the Behavioral Sciences journal. It is a pleasure to review your work.
I believe that the topic you have chosen is timely and crucial. Online aggression, particularly among college students (based on my personal experience), is a growing concern. With the increasing prevalence of social media and digital platforms, understanding the psychological mechanisms behind aggressive online behavior is more relevant than ever.
Aside from a timely and crucial topic, the papers have also several strengths that are commendable.
1. Sample size is large, which enhances the generalizability of the findings and adds robustness to your statistical analyses.
2. You made a good decision to use a mediation model. Your study goes beyond simple correlations to examine more complex interactions between variables
3. Your research contributes to the existing literature by extending the application of the general aggression model and fairness theory to online contexts, where anonymity and distance may impact aggressive behaviors.
Nevertheless, there are concerns with a few aspects of your paper that need clarification and refinement:
1. The connection between PRD, revenge motivation, and online aggression is central to your model, but the exact process through which PRD leads to revenge motivation could be clearer. How exactly does this sense of deprivation translate into the desire for revenge?
2. While it is hypothesized that violent attitudes moderate the relationship between PRD and online aggression, the explanation for how this moderation occurs is somewhat vague to me. I think it would be helpful to specify how violent attitudes interact with both revenge motivation and online aggression. Are there any nuances in this moderation process that you could further elaborate?
3. The cultural context in China is mentioned briefly, but more elaboration on how cultural factors specifically influence online aggression or attitudes toward violence would enrich the study. Could the findings be generalized to other cultural settings, or are they specific to the Chinese context? I suggest you address this area to make your paper more globally relevant or at least highlight the uniqueness of the findings.
4. I would try to expand, or at least highlight more, the real-world applicability of your practical implications. I believe offering more specific recommendations to develop intervention programs targeting revenge-driven online aggression will make your paper more citable.
That's it from me. Good luck with the revisions!
Reviewer 3 Report
Comments and Suggestions for Authors
This study developed a moderated mediation model that investigated the association between personal relative deprivation and online aggression. The overall writing is clear. Some suggestions are as follows:
Consider reconstruct the line 73 – 76. A lot of redundant use of words and citation missing.
Section 1.4 – would the author consider to add an hypothesis that test the relationship between PRD and online aggression directly.
Line 124-130 is not relevant!
Some more description might be needed for the scale of revenge motivation. What is the context for the sample question?
Formatting for table 2 is different from table 1.
Missing text in line 234-248. Perhaps these are for the description and insights of table 2 and figure 3?
“ICT model” for revenge motivation?? (Line 288)
The two limitations are similarly focusing on the self-reporting nature, would the authors provide a more reflexive thinking on the design or the problems in the research process?
Reviewer 4 Report
Comments and Suggestions for Authors
This study brings a new perspective on how revenge motivation mediates the link between personal private deprivation (PRD) and online aggression, and how a violent attitude moderates this connection, with a sample of 1004 college students Findings revealed a positive correlation between PRD and online aggression, with revenge motivation serving as a mediating factor, and, violent attitude moderation the relationship, indicating that PRD had a stronger association with online aggression in individuals with higher violent attitudes compared to those with lower attitudes.
The study is scientifically relevant, well-written, and grounded, and I believe it should be accepted for publication after minor revisions.
Lines 44-46 - The sentence is confusing. "Therefore, it is necessary to explore the influencing factors and psychological mechanism of college students' online aggression for the prevention and intervention in college students' online aggression". Do you mean "...necessary to explore possible influencing factor and psychological mechanisms of college students' associated with online aggression, to prevent and intervene in this severe behaviour"?
Line 104 - do you mean "Violent attitudes" (instead of Violence)
Materials and Methods - I believe the first paragraph was placed wrongly (it seems to be the instructions for this section...)
Participants: How were the participants recruited and were they all Chinese (nationality)? (it is not clear)
Measures. I think it is unnecessary to state that each scale "has been used in Chinese participants and has shown good reliability and validity". At the beginning of this section, the authors could state that "All scales have been used for Chinese participants and have shown good reliability and validity. Additionally, I believe it is unworthy to state that all scales reflect excellent reliability since the alpha value tells that itself. I would suggest putting the alphas inside parentheses at the end of the description of each variable.
Preliminary analysis: "The findings revealed that males reported higher levels of PRD, more violent attitudes, and greater online aggression than females, with significant correlations noted." I believe it would be more worthy to make this conclusion if a different means test was applied, rather than concluding this from correlations...
Reviewer 5 Report
Comments and Suggestions for Authors
This is a very interesting piece of paper. It is methodologically well planned and executed.
There are a number of suggestions for changes to the introduction (research hypothesis), methods, results and discussion sections.
In particular, the proposal to include a comparison of means is left to the discretion of the authors. The different correlations found between the variables analysed may justify its inclusion.
The differences in the relationship between the variables according to the gender variable must be taken into account when considering how the relationship between the Personal Relative Deprivation variable and the Online Aggression variable is explained by the mediation of the Revenge Motivation variable.
That is, revenge motivation explains why a situation of personal relative deprivation implies a high probability of online aggression. But does this relationship occur with the same intensity in men and women?
And if the moderating variable of violent attitudes modifies the effect of the relationship between the variable of personal relative deprivation and the variable of online aggression, we could ask a similar question: Does this relationship occur with the same intensity for men and women?
Dear authors. This is a work that I liked. I find it very interesting. I have not been indifferent to reading it.
The use of model 5 is adequate if the gender variable does not indicate that it may be moderating the relationship between the other variables.
Let us consider the moderating role played by gender. In this case, among the models proposed by Hayes, model 73 may be of interest.
Another possibility.
Considering the two research hypotheses, an option that can simplify the work:
a) Use models 8 and 14, with the variable gender as a moderator in the mediation relationship.
b) b) Use model 3 to determine the influence of the variable gender in the role of moderator of the variable Violent attitudes in the relationship between the independent variable (Personal relative deprivation) and the dependent variable (Online aggression).
Not mentioning gender as a moderating variable is the other option. Thus, PROCESS Model 5 can answer the research questions.
But knowing that gender is a relevant variable, this would be a very important way of biasing the work.

Reviewer 6 Report
Comments and Suggestions for Authors
In general I find the research relevant, interesting and I appreciate the high quality of your presentation of the theory and results. Some point below:
1. Please add the first hypothesis about relationship between PRD and online aggression - you stated in the theory that it is not well-studied in previous research
2. Could you please describe in more details what forms of online aggression are measured in the scale you used? There are many different forms and it’s unclear what kind of behavior is studied. Also, I suggest to add this information to Discussion – for others forms of online aggression results could be different.
3. It’s surprising that you used SEM from Mplus for mediation but then PROCESS for moderation. If there are some reasons for that probably you could add them. This point is up to you, just a commentary about the whole perception of this Data Analysis section
Round 2
Reviewer 3 Report
Comments and Suggestions for Authors
Thank you for addressing most of my concerns. The later of this manuscript would need a moderate level of editing starting from section 3.3
Is section 6 title correct?
The reference #1/ #4 look strange as well. Please have a thorough check.
Reviewer 5 Report
Comments and Suggestions for Authors
Title: Personal Relative Deprivation and Online Aggression in Col-2 lege Students: A Moderated Mediation Model of Revenge Motivation and a Violent Attitude
Dear authors, thank you for your response. This is an interesting work. Technically well done. I think the changes you made are appropriate.
1. Indicated fragment has been removed
2. Participants are described (age and gender)
3. The analysis procedure is justified
4. The moderating effect of gender is included
